



# In situ measurements of desert dust particles above the western Mediterranean Sea with the balloon-borne Light Optical Aerosol Counter/sizer (LOAC) during the ChArMEx campaign of summer 2013

Jean-Baptiste Renard[1], François Dulac[2], Pierre Durand[3], Quentin Bourgeois[4], Cyrielle Denjean[5], Damien Vignelles[1], Benoit Couté[1], Matthieu Jeannot[1,a], Nicolas Verdier[6], Marc Mallet[3,b]

[1]Laboratoire de Physique et Chimie de l'Environnement et de l'Espace (LPC2E), UMR CNRS-Université d'Orléans, 3A avenue de la recherche scientifique, Orléans, France
[2]Laboratoire des Sciences du Climat et de l'Environnement (LSCE), UMR CEA-CNRS-UVSQ, IPSL, Université Paris-Saclay, CEA Saclay 701, Gif-sur-Yvette, France
[3]Laboratoire d'Aérologie, Université de Toulouse, CNRS, UT3, Toulouse, France
[4]Department of Meteorology and Bolin Centre for Climate Research, Stockholm University, Stockholm, Sweden
[5]Centre National de Recherches Météorologiques (CNRM), UMR Météo-France-CNRS, OMP, Météo-France, Toulouse, France
[6]Centre National d'Etudes Spatiales (CNES), 18 Avenue Edouard Belin, Toulouse, France
[a]now at MeteoModem company, Chemin du Moulin, Ury, France
[b]now at CNRM[5]

**Abstract.** Mineral dust from arid areas is a major component of the global aerosol and has strong interactions with climate and biogeochemistry. As part of the Chemistry-Aerosol Mediterranean Experiment (ChArMEx) to investigate atmospheric chemistry and its impacts in the Mediterranean region, an intensive field campaign was performed from mid-June to early August 2013 in the western basin including in situ balloon-borne aerosol measurements with the Light Optical Aerosol Counter (LOAC). LOAC is a counter/sizer that provides the aerosol concentrations in 19 size classes between 0.2 and 100 μm, and an indication of the nature of the particles based on dual angle scattering measurements. A total of 27 LOAC flights were conducted mainly from Minorca Island (Balearic Islands, Spain) but also from Ile du Levant off Hyères city (SE France) under 17 Light Dilatable Balloons (meteorological sounding balloons) and 10 Boundary Layer Pressurized Balloons (quasi-Lagrangian balloons). The purpose was to document the vertical extent of the plume and the time-evolution of the concentrations at constant altitude (air density) by in situ observations. LOAC measurements are in agreement with ground-based measurements (lidar, photometer), aircraft measurements (counters), and satellite measurements (CALIOP) in case of fair spatial and temporal coincidences. LOAC has often detected 3 modes in the dust particle volume size distributions fitted by lognormal laws at roughly 0.2, 4 and 30 μm in modal diameter. Particles larger than 40 μm were observed, with concentrations up to about $10^{-4}$ cm$^{-3}$. Such large particles were lifted several days before and their persistence after transport over long distances is in conflict with calculations of dust sedimentation. We did not observe any significant evolution of the size distribution during the transport from quasi-Lagrangian flights, even for the longest ones (~1 day). Finally, the presence of charged particles is inferred from the LOAC measurements and we speculate that electrical forces might counteract gravitational settling of the coarse particles.

## 1. Introduction

Mineral dust from arid and semi-arid areas is a major component of the global aerosol and has long been recognized to have strong interactions with climate and biogeochemistry (e.g., Buat-Ménard and Chesselet, 1979; Martin et al., 1991; Swap et al., 1992; Duce, 1995; Alpert et al., 1998;



Mahowald et al., 2009; Maher et al., 2010; Liu et al., 2011; Mahowald et al., 2011; Choobari et al., 2014; Li et al., 2016). Desert dust aerosol is of particular interest in the Mediterranean region where it is frequently observed in high concentrations in the troposphere, being a major component of surface $PM_{10}$ (Pey et al., 2013; Rea et al., 2015), aerosol optical depth (Moulin et al., 1998; Gkikkas et al., 2013; Nabat et al., 2013), atmospheric deposition (Pye, 1992; Vincent et al., 2016), affecting the regional air quality (Querol et al., 2009), atmospheric thermodynamics (e.g. Alpert et al., 1998; Chaboureau et al., 2011), radiative budget and climate (e.g., Nabat et al., 2012, 2015a, 2015b), precipitation chemistry (Chester et al., 1996; Loÿe-Pilot et al., 1986; Avila and Rodà, 2002), soil formation (Nihlén et al., 1995), and biogeochemistry of forest ecosystems (Avila and Peñuelas, 1999), oligotrophic lakes (Morales-Baquero et al., 2006; Reche et al., 2009) and marine surface waters (Guerzoni et al., 1999; Herut et al., 1999; Guieu et al., 2014).

Most studies to characterize airborne dust particles transported long-range were performed with satellite remote sensing and/or surface in-situ and remote sensing instruments (counters, particles samplers, lidars, photometers…). Some aircraft observations were also conducted in situ inside dust plumes, but they are expensive and scarce (e.g., Schmid et al., 2000; Dulac and Chazette, 2003; Reid et al., 2003a; Formenti et al., 2008; Weinzierl et al., 2009; Denjean et al., 2016). In particular, there is some debate in the literature on the very-long distance transport of coarse soil dust particles (>10 μm in diameter). It has been shown that a coarse mode at about 14 μm in diameter is produced by sandblasting of arid soils by saltating sand grains (Alfaro et al., 1998; Alfaro and Gomes, 2001). Furthermore, d'Almeida and Schütz (1983) report that African dust storm conditions produce a dust particle volume size distribution extending up to several tens of μm with a highly variable 'giant' mode around 60 μm in diameter. The modified Stokes-Einstein law indicates that steady state gravitational settling velocities ($V_g$) of particles in air are proportional to the squared particle diameter (Stokes, 1851). For a particle density of 2.5 g cm$^{-3}$ typical of soil dust, $V_g$ reaches 1 cm s$^{-1}$ between 11 and 12 μm and 10 cm s$^{-1}$ between 36 and 37 μm, i.e. 860 and 8600 m d$^{-1}$, respectively (Foret et al., 2006). Those giant particles are therefore expected to fall and control the dust deposition flux within the first 1000 km of transport from their source (Schütz et al., 1981). However, there are evidences of Aeolian dust transport and sedimentation in the ocean up to 10 000 km away from source regions in the tropical Atlantic (Prospero et al., 1970; Carder et al., 1986) and Pacific (e.g. Betzer et al., 1988; Middleton et al., 2001; Jeong et al., 2014). Thus, there was a need of a new strategy for multiplying in-situ measurements of the dust particle size distribution during the transport of dust plumes. This was done for the first time in this study during African dust transport events above the western Mediterranean, deploying optical particle counters both below sounding balloons that crossed vertically the dust plume, and aboard drifting balloons that remained at constant altitude for quasi-Lagrangian measurements within the atmospheric dust layer.

The Chemistry-Aerosol Mediterranean Experiment (ChArMEx; http://charmex.lsce.ipsl.fr) is an international research initiative to investigate atmospheric chemistry in the Mediterranean region and its impacts on air quality, marine biogeochemistry and the regional climate. Within the project, a large regional field campaign was performed from mid-June to early August 2013 with intensive airborne measurements including in situ balloon-borne aerosol (Mallet et al., 2016) and ozone (Gheusi et al., 2016) measurements. The observations were conducted during the dry season over the western and central Mediterranean basins. During the first special observation period (SOP) entitled Aerosol Direct Radiative Forcing on the Mediterranean Climate (ChArMEx/ADRIMED SOP-1A) from mid-June to early July, the focus was on aerosol-radiation measurements and their modelling (Mallet et al., 2016). During the second SOP entitled Secondary Aerosol Formation in the Mediterranean (ChArMEx/SAFMED SOP-1B) from mid-July to early August, the focus was on atmospheric chemistry (Zannoni et al., 2017).

The present paper focuses on balloon-borne measurements conducted over the western Mediterranean during desert dust episodes encountered during this summer campaign with the new Light Optical Aerosol Counter (LOAC), an optical particle counter/sizer (OPC). Renard et al. (2016a and b) present the LOAC instrument and preliminary results from some flights analysed here with more details. In the following, we first briefly summarize the instrument principle and performances



and we describe the different sounding and drifting balloon flights performed in summer 2013
(section 2). Results on the particle size-segregated dust concentration are then presented, first in
terms of vertical distribution (section 3), and second in terms of temporal evolution at constant
altitudes (section 4). We then discuss dust particle sedimentation aspects (section 5) and
speculations about electrically charged dust particles (section 6), and finally conclude (section 7).
**2. Experimental strategy**
This study is based on the LOAC instrument (Figure 1), a light OPC described and characterized by
Renard et al. (2016a). Briefly, the instrument provides aerosol particle concentration measurements
within 19 size classes in the 0.2–100 μm diameter size range, and an estimate of the typology of
aerosols based on dual angle measurements. LOAC can be carried by all kinds of balloons (Renard et
al., 2016b). The gondola weight, including the instrument, the batteries (alkaline or lithium) and the
telemetry system, is of about 1.0 kg, for an electric consumption of 3 W. Aerosols are sucked in by a
small pump in order to pass through a red laser diode beam. In general, the light scattered by the
particles depends on both the size and refractive index of the particles. To separate these two
parameters, LOAC uses an original concept described in Renard et al. (2016a). Measurements are
performed at 2 scattering angles: the first one is close to forward scattering at around 12° where the
light scattered (diffracted) by non-spherical particles is controlled by the size of the particles (Lurton
et al., 2014); the second one is around 60°, where the scattered light is strongly dependent on the
refractive index of the particles (e.g., Weiss-Wrana, 1983; Renard et al., 2010; Francis et al., 2011).
The 12° channel is used to retrieve the size distribution independently of the nature of the particles,
and the combination of the 12° and 60° channels is used to derive the "LOAC speciation index" that
informs on the typology or dominant nature of aerosol particles in each size range, based on a
laboratory calibration conducted with particles of well-known nature. Figure 2 presents the reference
"speciation zones" obtained in laboratory and an example of LOAC speciation index obtained during
ambient air measurements inside a Saharan dust plume on 18 June above Minorca (Spain) at an
altitude of 3.1 km.
As described by Renard et al. (2016a), the measurement uncertainty on the total aerosol
concentration is ±20% for concentration values greater than 1 particle per cm$^3$ (for a 10-min
integration time). In contrast, the uncertainty is up to about 60% for concentration values smaller
than $10^{-2}$ particle per cm$^3$. In addition, the uncertainty in size calibration is ±0.025 μm for particles
smaller than 0.6 μm, 5% for particles in the 0.7-2 μm range, and 10% for particles larger than 2 μm.
Following coincidences, the measurement accuracy for submicronic particles could be reduced in a
strongly turbid case when the concentration of particles larger than 3 μm exceeds a few particles per
cm$^3$.
During the ChArMEx summer 2013 campaign, the LOAC gondolas were carried by two types
of balloon: the Light Dilatable Balloon (LDB), a meteorological sounding balloon of about 1 kg, and
the Boundary Layer Pressurized Balloon (BLPB), a drifting balloon of about 2.5 m in diameter.
Pictures of the respective gondolas can be found in Renard et al. (2016b).
The LDB allows many flights from various places, and the gondola may generally be retrieved
after landing (if not at sea). Measurements were conducted during the ascending phase of the
balloon, at a speed of 3-6 m s$^{-1}$. The inlet that collects aerosols was oriented toward the sky. The low
flow rate (~1.7 L min$^{-1}$) of the sampling pump yields sub-isokinetic sampling conditions that could
tend to oversample large particles (Renard et al., 2016a). The highest altitude reached by LOAC was
37 km, although in this study we will only consider the tropospheric part below 8 km in altitude (see
Chane Ming et al. (2016) for an analysis of upper troposphere and stratosphere observations). The
LOAC measurements, integrated every 10 s, are sent to ground in real-time by the on-board
telemetry. To increase the measurement accuracy during the LDB ascent, the 10-s concentration
values are averaged over a 1-min period, which provides a vertical resolution of about 300 m.



The BLPB, after its ascending phase, follows a near-Lagrangian trajectory, remaining in the
same air mass during its trajectory in the lower atmosphere (Ethé et al. 2002; Gheusi et al., 2016;
Doerenbecher et al., 2016). Its float altitude was prescribed before the flight (in the 400-3500 m
range) by adjusting the balloon density with the appropriate mixture of air and helium. The altitude
was chosen to fly within dust layers, based on a LDB flight and/or aerosol lidar measurements
performed just before the launch. At the float level, the horizontal speed of the drifting balloon
relatively to ambient air is close to zero, thus the particle sampling efficiency should be close to
100%. The integration time was chosen between 1 and 20 min, due to the low telemetry rate for the
downlink through the Iridium satellite communication system. The duration of the flights varied from
several hours to more than one day. Also, LOAC was sometimes temporarily shut down after a
session of measurements to save up on-board energy. For safety reasons, the authorized flight area
was restricted to the sea (including islands).
Table 1 and Table 2 provide the conditions of measurements for LBD and BLPB flights
performed during the ChArMEx campaign, respectively. Seventeen LDB flights and 10 BLPB flights
were successfully performed during desert dust transport events, most of them launched from
Minorca, the easternmost Balearic Island, Spain (latitude 39.88°N, longitude 4.25°E) from 15 June to
2 July, and a few from Ile du Levant, off Hyères city near the coast of south eastern France (latitude
43.02°N, longitude 6.46°E) from 27 July to 4 August (Figure 3). Those dust events were identified by
near-real time (NRT) model and remote sensing products collected operationally by the ChArMEx
Operation Centre web server (http://choc.sedoo.fr) where quick-looks were available. The main NRT
remote sensing aerosol products were provided by 4-hourly observations from MSG/SEVIRI. The
aerosol optical depth (AOD) at 550 nm ($AOD_{550}$) product is based on Thieuleux et al. (2005). In
addition, we operated a calibrated ground-based CIMEL AERONET sun-photometer that provided
AOD at 7 wavelengths from 340 to 1020 nm during daytime at the nearby station of Cap d'en Font on
Minorca Island (39.826°N, 4.208°E; http://aeronet.gsfc.nasa.gov) where an aerosol and water vapour
Raman lidar (WALI) with polarisation measurements was also in continuous operation (Chazette et
al., 2016). The balloon launch site and the lidar and photometer station were distant by about 10 km.
The confirmation of the occurrence of mineral dust plumes was possible from the LOAC-derived
typology of aerosol particles with the LOAC speciation index falling inside the "mineral zone" (Renard
et al., 2016a and b).
In case of a strong dust event, the measurement strategy was to perform two LDB flights per
day, and two simultaneous BLPB flights drifting at different altitudes within the dust plume (twin
flights). The flight altitudes were chosen following real-time indications from the nearby lidar. This
strategy was conducted during a relatively long dust event from 15 to 19 June, with 9 LDB flights and
3 twin BLPB flights on June 15, 16, and 19. The MODIS satellite observations indicate that the mean
AOD was of about 0.25 during this period. The dust started to appear over the Alboran Sea on June
12. The daily average AOD derived from MSG/SEVIRI over the western Mediterranean basin from
June 15 to 18 is mapped in Figure 4. It shows the arrival of the plume from the South-West with a low
AOD over Minorca on June 15, its extension to the North and North-East on June 16 and 17 with a
maximum extent of the plume over the basin on June 17, its reinforcement along a North-South axis
on June 18 with the largest AOD values around the Balearic Islands. On June 19 (not shown) Minorca
was on the western edge of the plume that shifted eastward.
The WALI lidar provides the vertical extent, the time-evolution and an estimate of the nature
of the particles during this event. Figure 5 shows times series of products from the lidar from late
June 15 to the end of June 17. The high extinction areas below 2.5 km until June 16, 13:00 are not or
weekly depolarizing. Chazette et al. (2016; see their Figure 7) could infer from those data that the
dominant aerosol was of marine nature around 500 m in altitude within the atmospheric boundary
layer, dust over 2.5 km, and pollution-related in between during the night of 15 to 16 June.
Five LDB flights were also conducted during the 28 June-2 July period. On 27-29 June, the
Minorca region was affected by turbid air masses arriving from the North-West (Chazette et al.,
2016). Ancellet et al. (2016) identified long-range transport of forest fire smoke from different areas
in North America (Canada and Colorado) and of African dust back from the western tropical Atlantic.



Their Flexpart model simulations indicate that over Minorca, Canadian smoke aerosols dominated
below 3 km on June 28 late afternoon, when dust dominated above 4 km and Colorado smoke
aerosols were abundant above 5 km. Satellite-derived AOD shows that starting on June 29, a new
dust plume from northwestern Africa with high AOD emerged from the Atlantic and Mediterranean
coasts of Morocco. The plume extended a bit to the North and further East over the sea during the
following days but remained confined to the southernmost part of the basin, with moderate AODs
and some dust over Minorca on 2 July 2 (Chazette et al., 2016). A BLPB flight was conducted on 2
July; the mean MODIS AOD was of 0.15 during this period. Lidar data indicate that dust dominated
between about 2 and 4.8 km in altitude (Chazette et al., 2016). The aerosol in lower layers could not
be typified.
Two other LDB flights were conducted from the Ile du Levant during a dust event on 27-28
July, with a mean MODIS AOD of 0.35. Finally, 2 LBD flights and one BLPB flight were conducted
during a last dust event on 3-4 August, with a mean MODIS AOD of 0.25.
The 27-28 June and the 2-3 July BLPB flights were the longest ones, with duration of about 1
day. Day-night transitions were thus encountered, leading to a decrease in float altitude during the
night of more than 100 m. Finally, the slow speed ascent of the 19 June and 3 August BLPBs allowed
us to obtain two additional fine-resolution vertical profiles in the lower troposphere.
From all those flights, it is possible to study the vertical extent of dust plumes and the
temporal evolution of the dust particle size-segregated concentration at a given altitude during
transport. In particular, LOAC data can be used to determine the concentration of large particles
dominating the mass of desert dust transported and their deposition flux (e.g. Arimoto et al., 1985;
Dulac et al., 1987, 1992a and b; Foret et al., 2006).

**3. Vertical profiles and particle size distributions of the observed dust plumes**

The main desert dust event observed during the ChArMEx/ADRIMED campaign lasted five days from
15 to 19 June as presented above (Figure 4). Figure 6 presents the vertical distribution of the 19 size
class number concentrations, from the 9 LDB flights performed during that period. It shows that the
dust plume was heterogeneously distributed in the free troposphere allowing for several local
concentration maxima along the vertical and extended up to 7 km on the evening of 18 June. For
comparison, Figure 7 presents measurements during a dust-free flight from Aire sur l'Adour, France
(43.706°N, -0.251°E) on 14 August 2014, with no significant local enhancement and the absence of
large particles.
All these flights, including a BLPB flight on 19 June morning when LOAC performed
measurements during the balloon ascent, were conducted concurrently to the nearby aerosol lidar
measurements (Chazette et al., 2016). The time of the first 5 LDB flights are presented on the WALI
lidar time-height cross-sections on June 16-17 with arrows marking. The LOAC aerosol number
concentration in the 0.2-100 μm range was converted to aerosol extinction using the Mie scattering
theory, assuming spherical dust particles, to be compared to the lidar extinction data at 350 nm. The
refractive index was set to 1.53-i0.0025, which corresponds to the mean value determined by
Denjean et al. (2016) for the Saharan dust plume events documented during the summer 2013
ChArMEx campaign. This approach suffers from three approximations: i) the contribution of the
smallest particles is unknown, leading to an underestimate of the calculated aerosol extinctions, ii)
the grains are considered as spherical while they are not and iii) the refractive index of the grains is
not always well known. In fact, the grains are irregular in shape and their refractive index can vary,
depending on their composition and their origin, which potentially increases the uncertainty on the
calculation of the aerosol extinction. Also, extinction calculations are highly sensitive to the size of
the particles: the uncertainty in the LOAC particle size determination can produce a 50% uncertainty
on the derived extinctions. Thus, the error bars on the LOAC-derived extinctions are calculated
considering both concentration and size uncertainties. Weinzierl et al. (2009) report that accounting
for the non-sphericity of dust particles might yield a small reduction of up to 5% in extinction





computations based on the dust particle size distribution. Figure 8 presents the tropospheric vertical
profiles of the LOAC and concurrent WALI lidar aerosol extinction observed during the 15-19 June
dust event over Minorca. Taking into account the uncertainties associated to the different
instruments, the overall aerosol extinction values can be regarded as in the same order of
magnitude, and even often in good agreement. LOAC and WALI have captured similar vertical
structures around half the time. The remaining discrepancies could be due to inaccurate size
determination by LOAC, and to the distance between different observations of inhomogeneous dust
plumes.
Three other dust events were documented with the LOAC instrument as illustrated in
Figure 9. The 28 June-2 July event was not very intense, while the 27-28 July and 3-4 August events
were stronger. Similarly to the mid-June event, the dust plumes extended up to an altitude of 6 km
and were not homogeneous in the vertical.
Good spatial and temporal coincidences occurred between LOAC measurements and Cloud-
Aerosol Lidar with Orthogonal Polarization (CALIOP on board the CALIPSO satellite) remote sensing
measurements for two events: on 29-30 June above Minorca (Spain) and on 3 August above Ile du
Levant (France). The LOAC measurements on 29-30 June were between 23:45 and 01:50 UT while the
CALIOP measurements were at 01:56 on 30 June. The LOAC measurements on 3 August were
between 11:15 and 12:15 while the CALIOP measurements were at 12:49. We used CALIOP version
4.10 level-2, 532 nm aerosol extinction data in the troposphere (e.g., Winker et al., 2009). The data
have a horizontal resolution of 5 km and a vertical resolution of 60 m. Aerosol extinction values have
a detection threshold of about 0.01 km$^{-1}$. The nature of aerosol particles and cloud droplets retrieved
in CALIOP observations is given by the CALIOP vertical feature mask algorithm (Omar et al., 2009). To
perform the comparison with CALIOP aerosol extinction data, the LOAC aerosol extinctions are
calculated at 532 nm from the measured size distribution using the mineral dust refractive index as
presented above. Lidar WALI extinctions are also available for the 29 June at around 22:30.
Figure 10 presents the comparison between LOAC, CALIOP and WALI aerosol extinctions.
During the 29-30 June night, the 3 instruments show that the plume extended from the ground to an
altitude of 2.5 km. Although the general trend is in good agreement for the 3 instruments, local
discrepancies are present in the vertical extinction profiles, possibly due to the temporal and spatial
variability of the plume. LOAC seems to indicate a mixture of mineral dust and carbonaceous
particles whereas CALIOP reports polluted continental/smoke particles (but the identification by
CALIOP is difficult due to the weakness of the signal). On 30 June at mid-day, the plume had almost
disappeared and the LOAC aerosol extinction values are below the detection threshold of CALIOP.
During the 3 August dust event, LOAC observations reveal that the plume extended from 2 to
6.5 km. CALIOP captured all the dust plume in very good agreement with LOAC, and the two
instruments identified the same nature of mineral dust particles. Another LOAC profile was obtained
in the morning of 3 August at about 06:30 UT, during a BLPB ascent up to its float altitude at 3 km (in
blue in Figure 10). The two LOAC measurements are in very good agreement in the 2-3 km altitude
range. Below 2 km, the 2 flight measurements show that the detected typologies are dominated by
carbonaceous particles (likely anthropogenic aerosols). The strong temporal variability in particle
concentrations below 2.2 km is therefore not related to the dust plume.
An ATR-42 aircraft flight was conducted close to Minorca (50 km apart) during dusty
conditions in the morning of 16 June, at the same time of two LOAC balloon flights (LBD and BLBP).
The aircraft probed the dust layer in the 2.5-4 km altitude range. The instrumentation installed on
board the aircraft is described in detail in Denjean et al. (2016). The aerosol size distribution was
determined from an optical particle counter GRIMM 1.129 (nominal size range 0.25-32 µm), an Ultra
High Sensitivity Aerosol Spectrometer (UHSAS; 0.04-1 µm) and a Forward Scattering Spectrometer
Probe FSSP-300 (0.28-20 µm). The UHSAS and FSSP are wing-mounted instruments, whereas the
GRIMM installed inside the cabin received ambient air collected through an isokinetic inlet and
tubing with a cut-off diameter of 12 µm (Denjean et al., 2016). Figure 11 presents the comparison of
the size distributions measured by the 2 LOACs and the 3 aircraft counters at the maximum
concentration level of the dust plume (2.5-4 km; see Figures 5 and 6). The integration from 2.5 to



4 km of the LDB LOAC signal provides a better signal to noise ratio and a better sensitivity to the less
numerous large particles (>15 μm) that are hardly detected with short integration times. Globally, all
the instruments are in good agreement for the submicronic particles and for the coarse mode at 2-
3 μm; the small discrepancies can be due to the difference in the respective measurement locations
and to the different measurement methods of the various instruments, although they were all
calibrated. The FSSP shows larger concentrations for particles larger than 2 μm in diameter than
other instruments (Figure 11). Reid et al. (2003b) discuss that the FSSP measurement principle tends
to produce some oversizing of coarse particles and also shows particle concentrations as high as
twice those measured by a Passive Cavity Aerosol Spectrometer Probe (PCASP) in their overlapping
particle diameter range (1.5-3 μm). This could explain such a shift in our dataset. It is worth noting
that the two LOACs, the FSSP and the GRIMM (despite the 12 μm cut-off of its sampling inlet) all
report particle concentrations larger than $10^{-3}$ cm$^{-3}$ around 20 μm in diameter. Both LOAC flights have
detected similar concentrations of particles in the channels larger than 22 μm in diameter. Although
the GRIMM counter on board the ATR42 aircraft could sense particles up to 32 μm, it did not report
such large grains, most probably because of the difficulty to collect and carry them up to the
instrument inside the aircraft cabin.
In Figure 12, the LOAC-derived size distributions were converted to volume concentrations
assuming spherical particles, using the mean volume diameter of each size class (Renard et al.,
2016a), and integrated over the whole vertical. The LOAC volume size distribution is compared to
that derived from the AERONET remote-sensing photometer. According to Dubovik and King (2000)
AERONET retrievals are limited to particles smaller than 30 μm and are not very sensitive to the
largest particles. Both LOAC and AERONET volume size distributions are in very good agreement in
the 0.2-20 μm size range, although the LOAC measurements show the presence of a third mode of
particles larger than 20 μm. Since the concentration of these large particles is low, the analysis of this
mode from measurements during LBD flights is limited. Long duration measurements performed at
constant altitude using the LOAC instrument on BLPB gondola with much longer measurement
integration time are better adapted to evaluate the concentration of these large particles and to
discuss this third, giant size mode.

## 4. Temporal evolution of the dust aerosol concentration and particle size distribution at constant altitudes

Figure 13 presents results from the BLPB flights performed inside dust plumes and Table 2 details the
conditions of measurements. Twin flights were performed on 16 June at the lower edge and in the
middle of the dust layer, on 17 June well inside the maximum concentration of the plume, and on 19
June at the minimum and maximum concentrations of the plume. On 27 and 28 June, flights were
performed in the upper edge of the plume. Finally, on 2 July and 3 August flights were conducted at
the maximum concentrations of the plume. It can be noticed that the 27 June and 2 July flights lasted
for about one day, with a day-night-day transition. In these cases, the altitude of the balloon is
slightly lower (from 100 to 200 m) at night than during the day due to the cooling of the balloon gas
and associated loss in buoyancy, so the night-time and daytime measurements were not conducted
in exactly the same air mass.
The LOAC aerosol concentration values obtained during the BLPB ascent on 19 June are in
good agreement with the LBD ascent measurements conducted at the same time (Figure 8, lower
right panel). In particular, LOAC has well captured the vertical variation of the dust plume
concentrations, with a local minimum at an altitude of 2 km.
During most of the flights, particles larger than 40 μm in diameter (last LOAC channel) were
detected. The concentration of these particles depends mainly on the intensity of the event, but the
highest concentration was detected in the free troposphere on 19 June, with $10^{-4}$ particles per cm$^3$ at



an altitude of 3.3 km. It can be noticed that concentrations of $10^{-3}$ particles per $cm^3$ were detected at
ground at the same date, as shown in Figure 6.
The dust particle volume size distributions were computed by integrating data over more
than a minute at a constant altitude. The mean diameter of the last channel was assumed to be 50
μm, although the size range is 40-100 μm, because the concentrations strongly decrease with size
and most of the particles thus have a diameter close to the lower limit of the size class. Those volume
distributions were then fitted with a 3-mode log-normal model using a least-square procedure. The
three fitted volume modal diameters ($Dm$) have been found at about 0.2, 4 and 30 μm as illustrated
in Figure 14. Note, however, that only the decreasing part of the first (small) mode is captured by
LOAC and the corresponding modal diameter could therefore be misestimated. There is also some
uncertainty on the third (large) mode related to the assumed upper limit of the measurement size
range and to the possible under-sampling of the upper tail of the size distribution; thus this mode
value may be a lower limit.
Figure 15 shows the evolution of $Dm$ for the three modes fitted from the 3 pairs of BPCL
LOAC data obtained during the 15-19 June dust event. BLPB flights lasted between 6 and 11 hours.
No significant temporal trend can be pointed out for $Dm$, meaning that the size distribution remains
almost constant over hours. Thus, it seems that no significant sedimentation has been detected
during the flights at quasi-constant altitude even for the very coarse mode at about 30 μm in
diameter.
Table 3 gives the values of $Dm$ for the 3 modes at float altitude for the 6 BPCL flights in dust
layers during the 16-19 June event; the average values are 0.26, 3.7 and 30.4 μm, respectively.
Values for the 3 modes are very comparable from one balloon to the other with a small variability of
about 15%, likely not significant given the uncertainties of the fitting. The flights inside the other dust
events confirm the presence of large particles in a giant mode at about 30 μm in diameter.
Nevertheless, some variations in aerosol concentrations often occurred. They are due to changes in
the balloon altitude during the day-night transition for the 27-28 June and the 2-3 July flights, to a
non-constant altitude for the 28 June flight, and to a slow ascent during the 3 July flight. These
variations of concentration are thus probably related to vertical variations of the dust plume layer.
Indeed, lidar profiles from Minorca show a strong vertical structuration of aerosol layers (Chazette et
al., 2016) that could be associated with significant differences in aerosol composition, concentration
and size distribution. Hamonou et al. (1999) first documented the multi-layered African dust
transport over the Mediterranean basin with variable source regions of mineral dust particles found
in different layers of the plume.
The presence of the third, very coarse mode with $Dm$ of the order of 30 μm may be related
to the existence of a mode in desert dust aerosols: a value of the same order ($Dm$ =42.3 μm) is
assumed in the model of background desert dust aerosol of Jaenicke (1987). Observations of a very
coarse mode are also reported by Weinzierl et al. (2009): particles larger than 20 μm were detected
in nine out ten cases of 49 pure dust layers observed in altitude over southern Morocco with wing-
mounted airborne optical particle counters. In 20% of the cases, particle sizes equal or larger than 40
μm and up to 80 μm were detected (with a detection limit of $10^{-2}$ $cm^{-3}$). They report an average
volume median diameter of the coarse mode of 15.5 ±10.9 μm near dust source region and a
maximum value larger than 60 μm in a case of strong convection. The better sensitivity of LOAC may
explain why we report more systematically a coarse mode of dust particles above 20 μm in diameter.
However, the persistence of such large particles, lifted several days ago, and their transport above
the Mediterranean basin is not well understood given their large theoretical settling velocity.
**5. Discussion related to dust sedimentation**
According to the Stoke's approximation that equates the effective weight of spherical
particles and the viscous resistance of the fluid through which it moves (Stokes, 1851), the





gravitational settling velocity $V_g$ of dust particles is proportional to the square of their diameter.
Assuming a classical density value of 2.5 g cm$^{-3}$ for spherical dust particles (Dulac et al., 1989; Zender
et al., 2003; Linke et al., 2006), $V_g$ is thus about 0.076, 0.19, 0.76, 3.0, 6.8, and 19 cm s$^{-1}$ for particles
of 1, 5, 10, 20, 30 and 50 μm in diameter, respectively, implying a downward transport ranging from
about 66 to 16 400 m d$^{-1}$. Particles larger than 12.3 μm have a sedimentation velocity larger than
1 000 m d$^{-1}$. This is supposed to yield a quick segregation and a rapid evolution of the dust particle
size distribution in the first days (and even hours) of transport after lifting from the dust source
region (e.g., Schütz et al., 1981; see also Figure 1 in Foret et al., 2006). Dust-loaded air masses
transported northward from Africa above the marine atmospheric boundary layer in the western
Mediterranean are known to be associated with warm fronts and to experience a significant upward
synoptic movement (Prodi and Fea, 1979; Reiff et al., 1986). Dulac et al. (1992a and b) report that
during a typical summer dust episode the turbid air mass ascending vertical velocity was on average
of the order of 1.5 to 1.8 cm s$^{-1}$ during 4 days, i.e. was more than compensating the average
deposition velocity of the bimodal dust particle size distribution observed in Corsica during this
event, with 2 modes at 2 and 13 μm. This is not enough, however, to explain the relatively constant
dust particle size distribution observed during BPCL flights: accounting for an average upward air
mass vertical velocity of 1.5 cm s$^{-1}$ that would counteract gravitation, a 4-km thick dust layer should
anyway lose by sedimentation all particles larger than 30 μm in about 1 day.
According to Slinn (1983; eq. 160), the flux-mean deposition velocity $(<V_d>)$ of a lognormal
distribution of particles of modal diameter $Dm$ and geometric standard deviation $\sigma_g$ can be derived
from $<V_d> = V_d(Dm)\ \sigma_g^{2\mathrm{Ln}(\sigma g)}$. Using this formula, we can derive that the 3-fitted dust particle size
modes shown in Figure 14 have a respective gravitational settling velocity of about 0.0011, 0.50, and
8.1 cm s$^{-1}$, corresponding to a negligible downward transport by sedimentation of about 1 m d$^{-1}$ for
the finest mode, but to 430 m d$^{-1}$ for the intermediate mode, and as much as 7 000 m d$^{-1}$ for the
largest one. Figure 13 does not show any significant systematic evolution of the concentration of the
different modes. New particles sedimenting from turbid layers above the balloon might compensate
for the sedimentation of particles from the intermediate coarse mode with $Dm$ of about 4 μm during
our 1-d or less balloon flight times. However, we should definitely observe a significant decrease in
the concentration and median size of the very coarse mode with $Dm \approx 30$ μm. Figure 15 does not
show any evidence of a decrease in the very coarse mode median diameter.

Laboratory tests have shown that the LOAC photodiode and electronics are sensitive to
electromagnetic fields, as those generated by radio telemetry, by strong atmospheric electric activity
(e.g., during thunderstorms), and even by an electrical bay. In these cases, the electronic noise and
the electronic offset both increase. The offset also increases with increasing ambient temperature.
LOAC performs measurements of the electronic noise and offset every 15 min when the light
source is switched off (Renard et al., 2016a). During a typical LDB flight, the LOAC electronic offset
slightly decreased with altitude due to the decreasing temperature encountered during the balloon
ascent. In contrast, an offset increase correlated to the increase in dust particle concentration was
detected for 5 flights inside a dust plume (Figure 16). It seems that increases in the amplitude of the
offset and in concentration were correlated. No conclusion can be derived from the other flights
inside the plumes, since no offset control was done close to the maximum in dust concentration.
These offset increases may be related to the presence of local strong electromagnetic fields inside
the plume, although it is not possible to retrieve their strength with such kinds of measurements. It is
known that the aerosol generation from both mineral dust powders (e.g., Johnston et al., 1987;
Forsyth et al., 1998) and arid soils (e.g. Ette, 1971; Farrell et al., 2004; Sow et al., 2011) produces
charged particles, and that electrical charges in sandstorms perturb telecommunication
transmissions (e.g. Li et al., 2010 and references therein). The presence of electric field in dust

## 6. Indirect detection of possible charged particles



aerosol layers was indeed proposed by Ulanowski et al. (2007) to explain the alignment of non-
spherical particles and polarization effects in a dust plume over the Canary Islands. Nicoll et al. (2011)
also report charged particles within Saharan dust layers with two balloon soundings performed
above Cape Verde Islands.
We suggest that electric forces within the dust layers could contribute maintaining in
levitation coarse particles that would otherwise expected to sediment down. Future balloon
campaigns with LOAC measurement in parallel with an adequate instrument retrieving accurately the
atmospheric electric field could consolidate these previous studies. This looks as an important
perspective to consider since the local electric field in dust plume might be at least partly responsible
for the non-sedimentation of large particles resulting in much longer transport than expected.
**7. Conclusions**
The in-situ LOAC balloon-borne measurements above the Mediterranean basin in summer 2013 have
allowed us to document both the vertical extent of the dust plumes and, for the first time to our
knowledge, the time-evolution of dust concentrations from several hours to one day in a quasi-
Lagrangian way at constant altitude. Whenever possible, LOAC observations were compared to the
measurements done by other platforms, like the ATR-42 aircraft which embarked various aerosol
counters, and a backscattering lidar located close to the balloon launching area. Given the limits and
uncertainties associated with each measurement system, the agreement was satisfactory, which
gave us confidence in the LOAC aerosol distributions. LOAC has often detected the presence of
particles larger than 40 $\mu$m, with concentrations up to $10^{-4}$ particles per $cm^3$, and the fitting of
volume size distribution ended up in a coarse mode at 3-4 $\mu$m in diameter and a giant mode at about
30 $\mu$m. Such large particles should have been lifted several days before, and at least 1 000 km far
from our measurements. Their transport over such long distances, not expected from calculations of
dust particle sedimentation, is yet not well understood. Indeed, the gravitational settling velocity of
dust particles between 12 and 40 $\mu$m in diameter spans from almost 1 to more than 10 km per day.
An indirect evidence of the presence of charged particles has been derived from the LOAC
measurements and we therefore hypothesize that electric forces within the dust plume might limit
the sedimentation of the coarse dust fraction.
ChArMEx was a unique experiment involving a large set of ground-based and airborne
instruments. Since 2014, regular LDBs with LOAC are launched twice per month from Aire-sur-l'Adour
(South-West of France, 43.71°N, 0.25°W) to monitor the aerosol content from the troposphere to the
stratosphere. Dust events were already occasionally detected; they will be used to document other
dust events than those of summer 2013 with the same instrument, and to confirm the presence of
both the large particles and the charged particles thanks to new developments of the instrumental
payload.

**Acknowledgments**. The LOAC development project was funded by the French National Research
Agency's ANR ECOTECH. The balloon flights of the MISTRALS/ChArMEx campaign were funded and
performed by the French Space Agency CNES. The LOAC instruments are built by Environment-SA
company; the balloon-borne gondolas are provided by MeteoModem company and by CNES for
sounding and drifting balloons, respectively. The numerous LOAC instruments used during the
campaign and the scientific and technical staff missions were funded with the support of CNES, INSU-
CNRS and ADEME. The SAFIRE team is acknowledged for the aircraft operation. The French
VOLTAIRE-LOAC Labex (Laboratoire d'Excellence ANR-10-LABX-100-01) also provided a couple of
LOAC instruments. The PHOTONS (http://loaphotons.univ-lille1.fr/) and AERONET
(https://aeronet.gsfc.nasa.gov/) teams are acknowledged for our sun-photometer calibration and
data processing, respectively. P. Chazette and J. Totems are acknowledged for lidar data from
Minorca. ChArMEx LOAC balloon and other measurements are available on the ChArMEx database





(http://mistrals.sedoo.fr/ChArMEx/). Finally, CALIOP data have been retrieved through the ICARE
Data Services and Center (http://www.icare.univ-lille1.fr).

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




| Date (2013) | Time (UT) | | Altitude range (km) | Launch site |
|---|---|---|---|---|
| | start | end | | |
| 15 June | 22:12 | 22:48 | 0.9 – 6.9 | Cap d'en Font, Minorca Isl., Spain (39.88°N, 4.25°E) |
| 16 June | 10:37 | 11:14 | 2.0 – 12.1 | |
| 16 June | 21:17 | 21:59 | 0.2 – 10.1 | |
| 17 June | 10:02 | 10:41 | 0.1 – 11.4 | |
| 17 June | 18:29 | 20:33 | 0.9 – 33.3 | |
| 18 June | 16:35 | 18:41 | 0.2 – 35.4 | |
| 18 June | 21:19 | 22:39 | 0.4 – 21.5 | |
| 19 June | 10:15 | 12:03 | 0.8 – 30.7 | |
| 19 June | 13:50 | 15:03 | 0.3 – 20.7 | |
| 28 June | 05:38 | 07:54 | 0.6 – 36.0 | |
| 29-30 June | 23:31 | 01:49 | 0.2 – 35.9 | |
| 30 June | 14:03 | 15:46 | 0.1 – 26.8 | |
| 2 July | 10:30 | 12:24 | 0.7 – 32.8 | |
| 27-28 July | 23:13 | 01:17 | 0.3 – 33.5 | Levant Isl., France (43.02°N, 6.46°E) |
| 28 July | 15:31 | 18:06 | 0.3 – 33.3 | |
| 3 August | 11:04 | 12:35 | 0.3 – 21.7 | |
| 4 August | 15:32 | 17:36 | 0.2 – 32.2 | |

**Table 1**. List of the seventeen LOAC flights under Light Dilatable Balloons (LDB) flown during African dust plume events of the ChArMEx summer 2013 campaign.




| Balloon # | Date (2013) | Time slot of LOAC data (UT) | | Drift altitude (km) | Latitude, longitude at end of flight | Flight length (km) | Ceiling duration (h) |
|---|---|---|---|---|---|---|---|
| | | start | end | | | | |
| B74 | 16 June | 10:00 | 21:28 | 2.1 | 42.892°N, 05.229°E | 361 | 11.3 |
| B70 | 16 June | 09:51 | 23:01 | 3.1 | 40.182°N, 06.128°E | 164 | 12.6 |
| B75 | 17 June | 09:31 | 16:23 | 2.0 | 42.815°N, 03.811°E | 362 | 6.4 |
| B72 | 17 June | 17:11 | 18:59 | 2.75 | 43.179°N, 04.800°E | 377 | 7.0 |
| B77 | 19 June | 10:25 | 16:54 | 2.55 | 43.042°N, 04.833°E | 369 | 6.0 |
| B71 | 19 June | 10:29 | 15:58 | 3.3 | 43.041°N, 05.151°E | 363 | 3.6 |
| B80 | 27-28 June | 09:50 | 12:31 | 3.0 | 37.916°N, 12.145°E | 719 | 25.3 |
| B73 | 28 June | 05:25 | 16:42 | 2.7 | 37.523°N, 08.830°E | 512 | 11.2 |
| B76 | 2-3 July | 13:04 | 09:14 | 3.2 | 37.880°N, 12.109°E | 717 | 19.3 |
| B82 | 3 August | 06:12 | 08:12 | 3.0 | 43.077°N, 06.662°E | 45 | 1.4 |


**Table 2**. List of the ten LOAC flights aboard drifting Boundary Layer Pressurized Balloons (BLPBs)
flown during African dust plume events of the ChArMEx summer 2013 campaign. All balloons were
launched from Minorca Isl. (Spain; 39.864°N, 4.255°N) except B82 that was launched from the Levant
Isl. (France; 43.022°N, 6.460°E).






| Date (2013) | Altitude (km) | Average volume median diameter (*Dm*, μm) ± standard deviation (*number of measurements*) | | |
|---|---|---|---|---|
| | | Mode 1 | Mode 2 | Mode 3 |
| 16 June | 2.1 | 0.22 ±0.02 (*32*) | 3.6 ±0.8 (*32*) | 30.6 ±3.4 (*32*) |
| | 3.1 | 0.30 ±0.07 (*31*) | 3.3 ±0.3 (*31*) | 28.5 ±1.7 (*30*) |
| 17 June | 2.0 | 0.26 ±0.02 (*17*) | 4.1 ±0. 6 (*17*) | 27.4 ±4.1 (*17*) |
| | 2.8 | 0.24 ±0.02 (*16*) | 3.3 ±0.5 (*16*) | 30.9 ±5.9 (*16*) |
| 19 June | 2.6 | 0.25 ±0.01 (*28*) | 3.5 ±0.6 (*28*) | 32.8 ±4.7 (*27*) |
| | 3.3 | 0.26 ±0.01 (*48*) | 4.5 ±0.5 (*48*) | 32.4 ±4.2 (*41*) |
| *Average* | | *0.26 ± 0.04* | *3.7 ± 0.4* | *30.4 ± 2.8* |

**Table 3**. Average volume median diameter (*Dm*) of the three fitted aerosol particle modes and
respective standard deviation along BPCL flights at float altitude within dust layers, for the 6 BPCLs
launched from Minorca during the 16-19 June 2013 dust event. The time evolution for the three pairs
of BPCL flights flown during this period is shown in Figure 15. The average and standard deviation in
the bottom line are obtained by averaging the 6 above values.






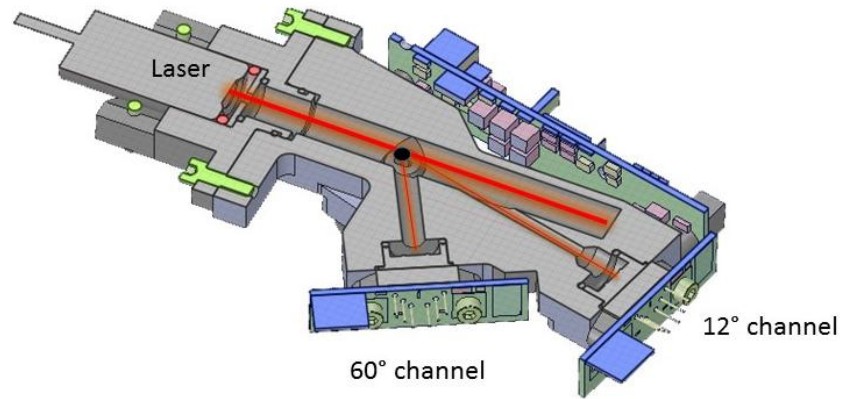

**Figure 1.** The LOAC instrument and principle of scattering measurements at two angles.






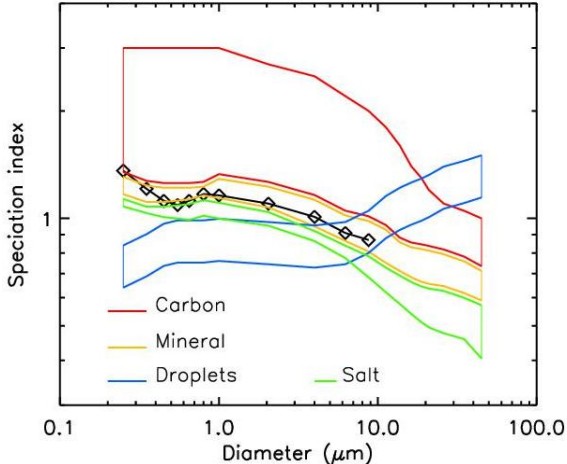

**Figure 2.** "Speciation zones" obtained in laboratory for several types of particles (color lines) and an
example of LOAC speciation index obtained during ambient air measurements inside a Saharan dust
plume at an altitude of 3.1 km (18 June 2013, 18:15 UT) above Minorca, Spain (diamonds).






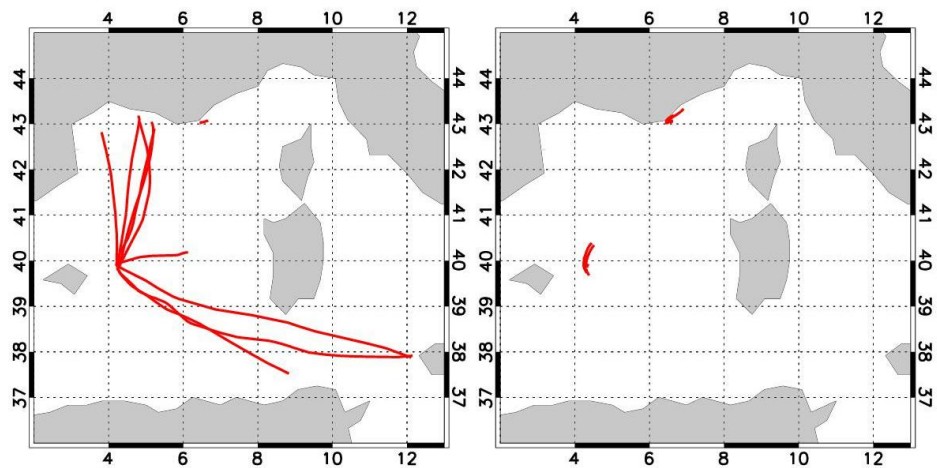

**Figure 3.** Trajectories of the flights for the 10 BLPBs (left) and 17 LDBs (right).






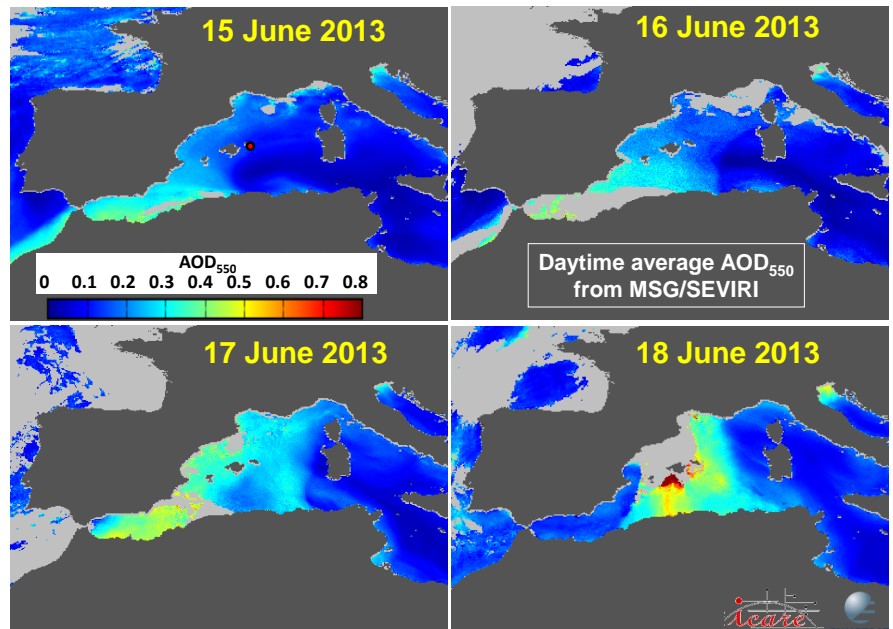



**Figure 4.** Daytime averaged Meteosat-derived AOD at 550 nm over seawater from June 15 to 18,
2013, showing the synoptic development of the dust event. The product is computed by the ICARE
data and services center (http://www.icare.univ-lille1.fr) based on the algorithm of Thieuleux et al.
(2005). Lands are masked in dark grey and clouds over ocean in light grey. The red dot on the 15 June
image indicates the balloon launching site and remote sensing station on Minorca Island.


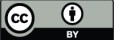




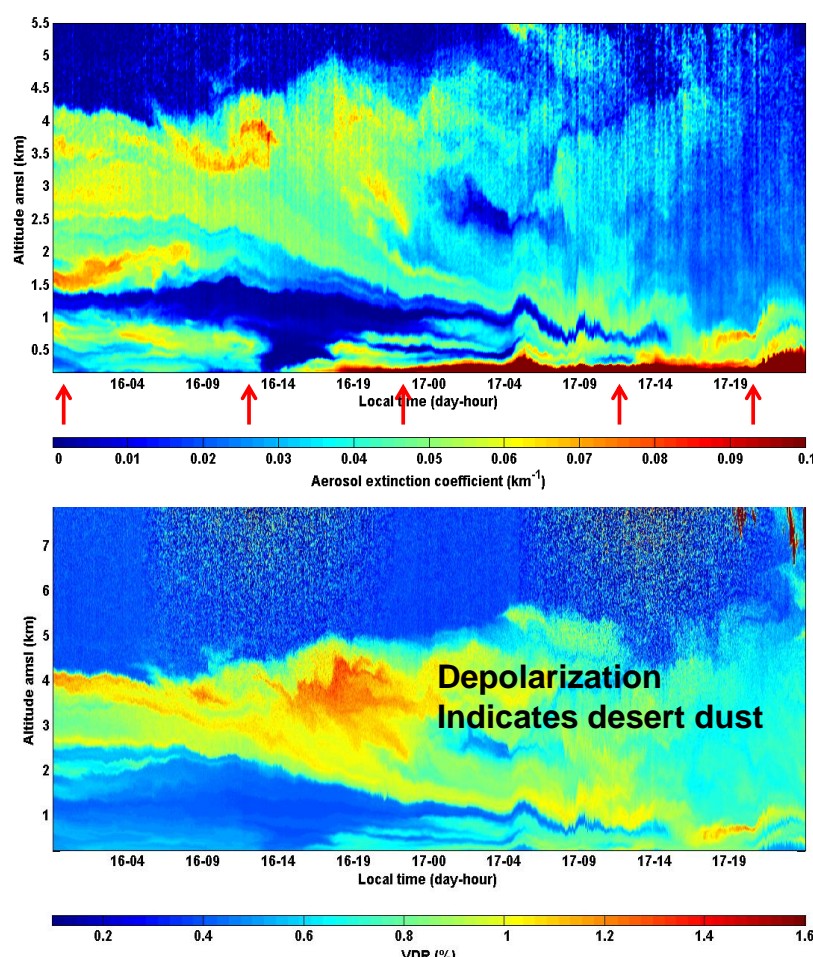

**Figure 5.** Lidar-derived time-height cross-sections of the aerosol extinction (top) and volume
depolarization ratio (bottom) at Minorca from June 15, 22:00 to 17, 24:00 UT. The red arrows
indicate the time of the 5 LDB launches. Courtesy of P. Chazette and J. Totems, after Chazette et al.
906 (2016).




**Figure 6.** Evolution of the dust plume from LOAC balloon measurements over Minorca, Spain, from
15 to 19 June. The ascent from 0 to 8 km takes about 30 min and the reported times of measurement
are taken at the middle of the profile.






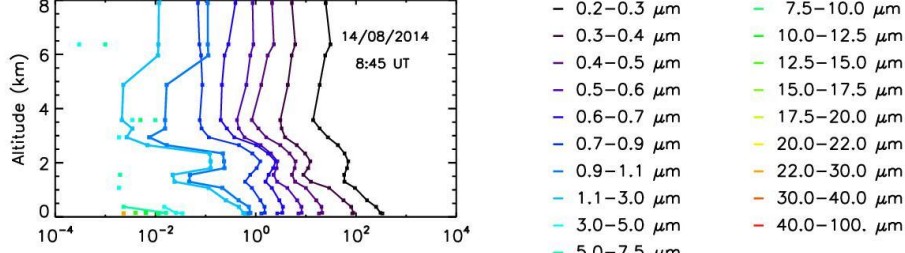

**Figure 7.** Typical vertical profile when no dust is present; flight from Aire sur l'Adour (South West of
France), on 14 August 2014.







**Figure 8.** Comparison between LOAC extinctions and WALI lidar extinction at 350 nm. The LOAC error bars consider the uncertainty on the LOAC measurements and on the counting-extinction conversion; the WALI error bars are calculated from the individual measurement scatter.



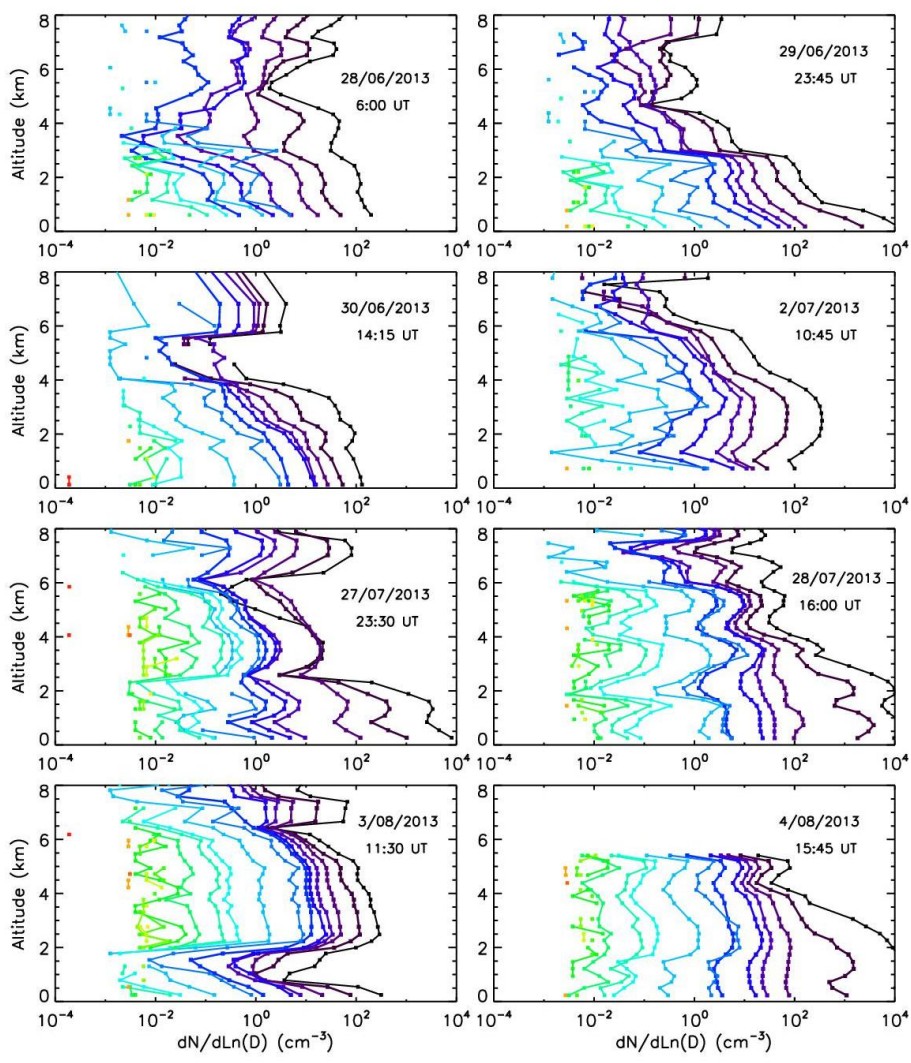

**Figure 9.** Same as Fig. 6 but for other sand plume events observed over Minorca (27 June – 2 July)
and Ile du Levant (27 July - 4 August).




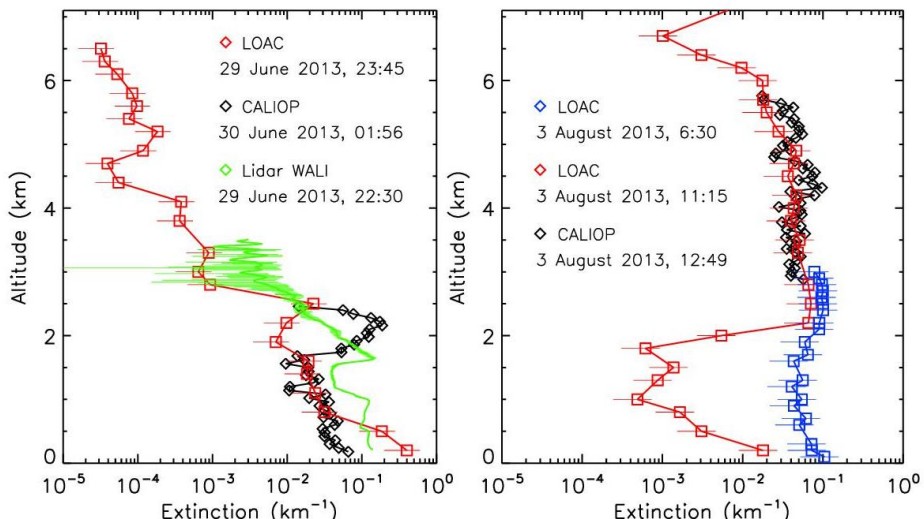

**Figure 10.** Left, vertical profiles of aerosol extinction from LOAC, CALIOP and WALI for the 29-30 June
event, above Minorca; right, vertical profiles of aerosol extinction from LOAC (LDB and BLPB flights)
and CALIOP for the 3 August event above Ile du Levant.






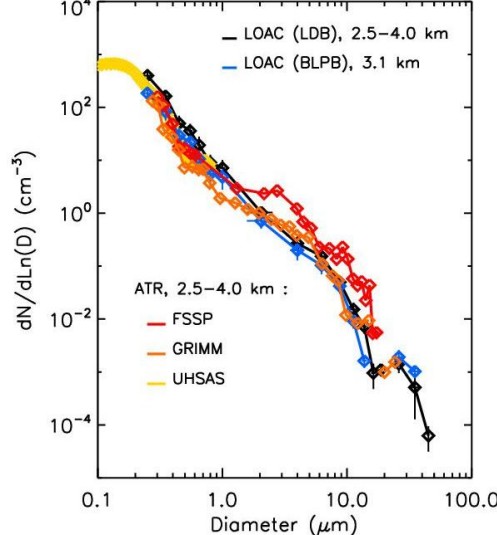



**Figure 11.** Comparison of the particle size distributions during the 16 June dust plume event over
Minorca, obtained with LOAC instruments under balloons and particle counters on board the ATR-42
aircraft; measurement altitudes are given.





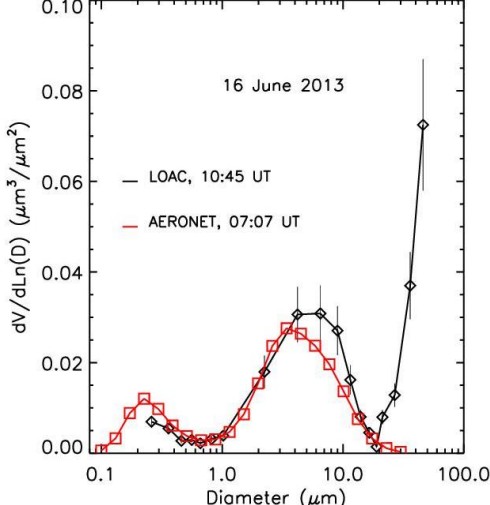

**Figure 12.** Volume size distribution retrieved from AERONET (https://aeronet.gsfc.nasa.gov/) and
LOAC data on 16 June at Minorca.





**Figure 13.** Flight altitude (up) and time evolution (bottom) of the LOAC-derived aerosol concentration
for the 19 size classes, from BLPB flights from Minorca towards French coast. Colour coding is as in
Fig. 6. Day-night transitions are indicated when appropriate.






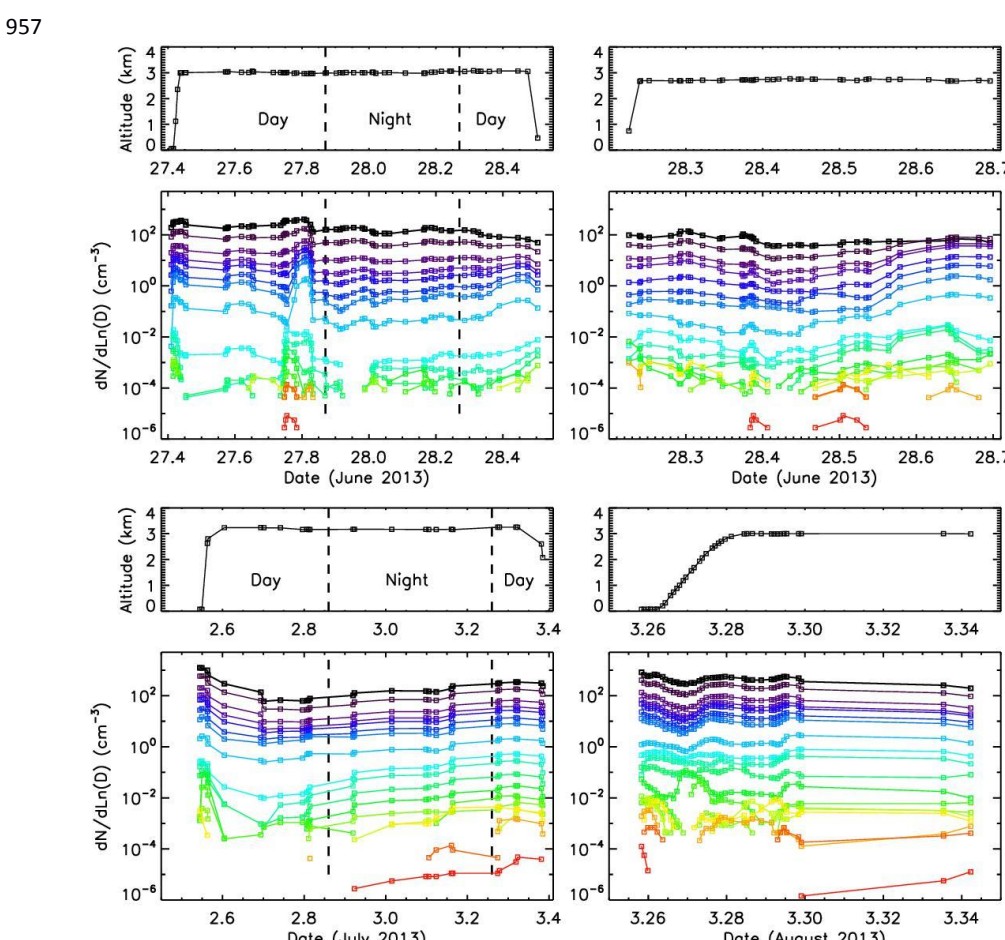

**Figure 13, continued.**





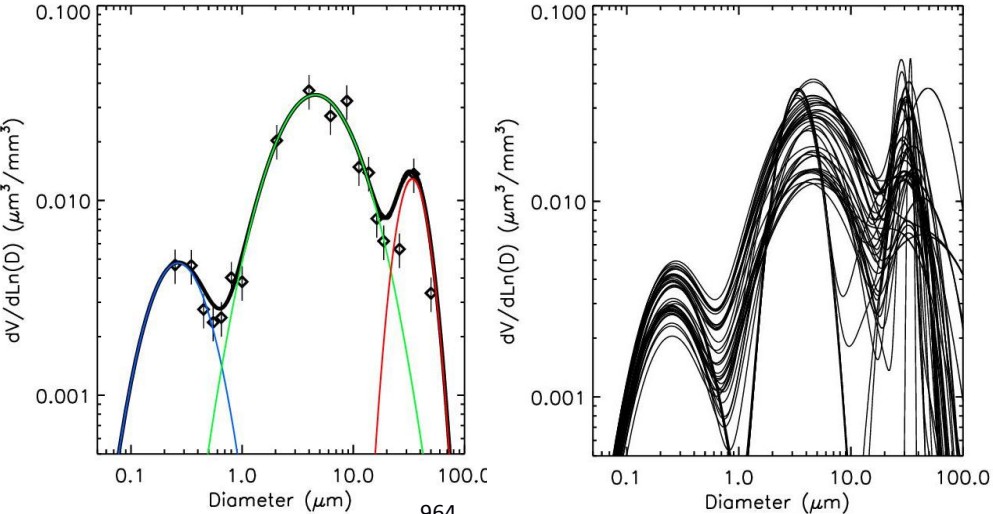

**Figure 14.** Left: Particle volume size distribution within the desert dust plume from the BLPB of 19
June at an altitude of 3.3 km (12:30 UT). The black diamonds are the LOAC measurements, the
coloured curves represent the lognormal functions for each of the observed modes, and the black
curve represents the overall fit (sum of the 3 modes). The geometric mean diameters (*Dm*) of the 3
modes are of 0.27, 4.6 and 34 μm, respectively, with respective geometric standard deviations (*σ*) of
1.79, 2.14 and 1.35. Right: The 41 fitted size distributions when the third mode was detected,
retrieved from measurements at BLPB float altitude.



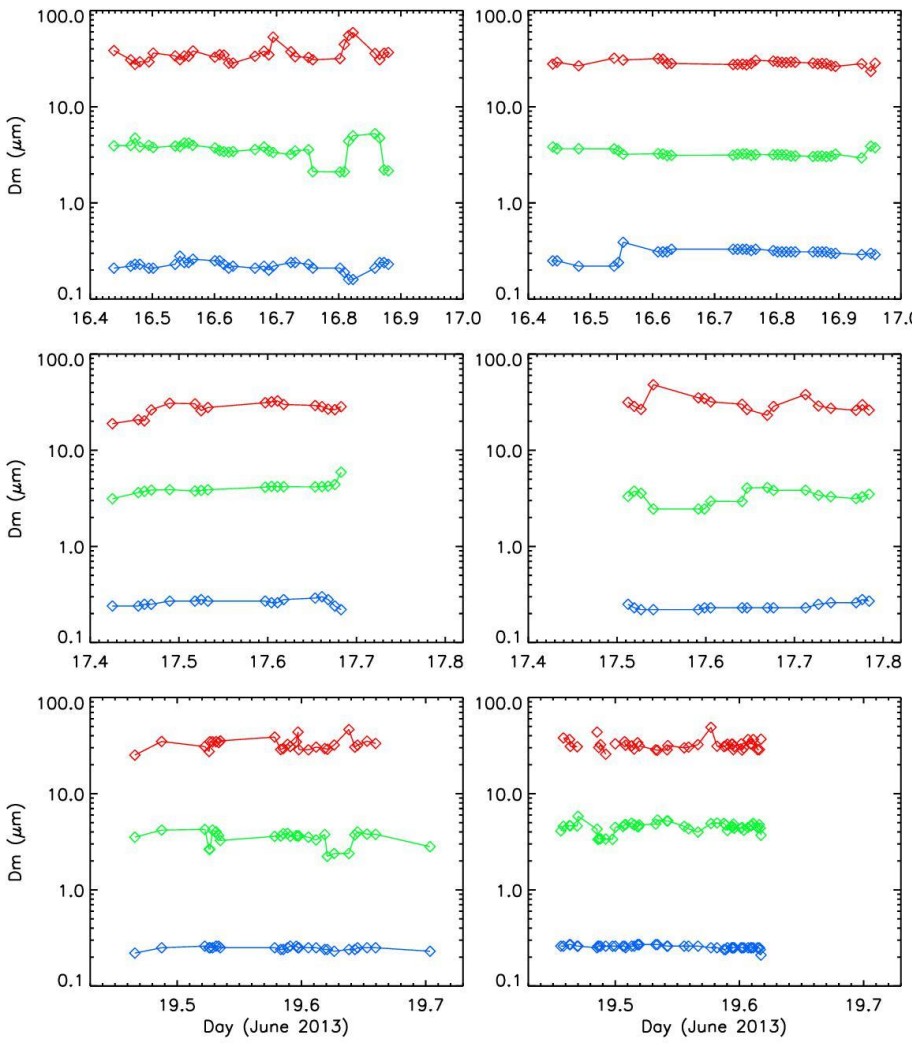

**Figure 15.** Time-evolution of the particle size at the maximum concentration for each mode (*Dm*) of the volume size distribution, at float altitude of BLPB flights from Minorca towards French coast. The altitudes are 2.1 and 3.1 km for the 16 June flights (top left and right, respectively), 2.0 and 2.7 km for the 17 June flights (middle left and right, resp.), and 2.5 and 3.3 km for the 19 June flights (bottom left and right, resp.). Average *Dm* values of the 3 modes during each flight are given in Table 3.





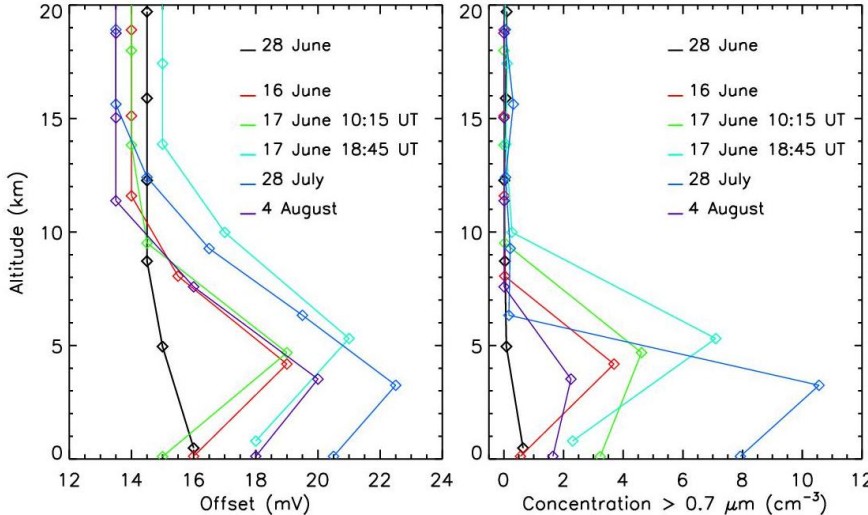

**Figure 16.** Left: Profiles of the LOAC electronic offset in case of crossing a strong dust plume (16 June,
17 June mid-time and evening, 28 July and 4 August) and in case of a weak dust plume just close to
ground on 28 June. Right: Profiles of number concentrations of dust particles larger than 0.7 μm for
the same flights.