# Peer review of "In situ measurements of desert dust particles above the western"

_Atmospheric Chemistry and Physics, 2017_

## Referee Comment (RC1) · Anonymous Referee #1 · 2 Dec 2017

The paper by Renard et al., presents in situ balloon-borne aerosol measurements with the Light Optical Aerosol Counter (LOAC) in dust layers over western Mediterranean. Measurements were performed either during the ascending phase of the balloon or in a quasi Lagrangian way at constant altitude and were compared to in-situ airborne and remote sensing measurements. The observations presented are important given the scarcity of similar observations within dust layers, and should be published after some minor revisions.

The technical descriptions of measurements are scattered throughout the manuscript (ie LIDAR, airborne, satellite) . It is strongly recommended that all technical details are given in a separate section described as material and methods, preferably before 2.Experimental strategy that could be a subsection. This will allow the reader to focus on the observations rather the technical details.

Specific comments:

Line 213:"July 2"

Line 244: Refer to the corresponding Figure 5.

Line 327: In this paragraph comparison to AERONET is briefly discussed and in Figure 12 a single measurement is presented to support the statement that LOAC and AERONET are in very good agreement. More data need to be presented, there must be several AERONET profiles available to compare. These data will be useful to the AERONET comunity as well, since these measurements may provide validation data for the inversion algorithms. Otherwise, that statement should be limited to a single day that good agreement was observed.

Line 344: This paragraph should be incorporated in Experimental strategy section, you are describing once again the flight patterns here.

Line 454: Evidence has to be given that this correlation exists otherwise this is rather a speculation and the sentence has either to be rephrased as a hypothesis or removed.

Figure 5. The same height resolution should be used, it is easier for the reader to compare the two plots.

Figure 11: Although the logarithmic scale shows good agreement between various instruments, it would be better to present concentrations in linear scale.

---

## Referee Comment (RC2) · Anonymous Referee #2 · 17 Dec 2017

This work summarizes the size distribution measurements conducted airborne using balloons during summer 2013. Even though 27 flights were conducted, the authors focus on a dust episode that occurred between 15 to 18 June 2013, which involved about half of these flights, because it was the most intense observed during the sampling period.

This work reports that particles greater than 20 $\mu$m were identified inside the dust

plumes, which is against expected. This is an important finding, within the scope of this journal that certainly should be published after a few minor changes are applied. On top of that, the article is very well written, and easy to follow through. The authors have tried, and have done a very good job, to discuss any questions raised while reading the article.

My main concern is the counting statistics of the LOAC at sizes greater than 40 $\mu$m. The authors report 1e-4/ccm as the maximum observed concentration. For 1 minute sampling (averaged over 6x10 sec measurements) in 1.7 Lpm means that 0.17 particles were sampled. Redard et al. (2016) reports that for the smaller size classes at least 400 particles must be detected to have a proper signal, which translates to 1 mV accuracy in the case of 20 mV noise. However, for higher channels it is not mentioned how many particles are required for a proper signal. Furthermore, Redard et al. (2016) reports a good match with a fog monitor over the size range in question, but for concentrations up to 0.1/ccm (ie 3 orders of magnitude higher).

As a result I encourage the authors to discuss the detection limit of LOAC at sizes greater than 40 $\mu$m. Additionally Redard et al. (2016) reports $\pm$60 % uncertainty for concentrations smaller than 1e$-$2/ccm. Since the uncertainty increases with decreasing concentration, can it be that in the 1e-4 range the uncertainty reaches $\pm$100 %?

My second major point has to do with the sampling. Two sampling methods were implemented, LDB and BLDB. From reading the article, I am left with the impression that both methods apply a vertical (to the ground) inlet. For LDB this is understood, but for the BLDB method, this would be devastating taking into account that inside dust plumes high wind speeds are frequently encountered. Can you please clarify?

Fig 14 is very puzzling to me. If I understand correctly diamonds (left graph) are the measurements and the vertical line on these diamonds the uncertainty (1 std? it is not mentioned). If this is the case the uncertainty line should overlap with 68% of the lines in the graph on the right. In other words the measurement uncertainty should

somewhat match the range of fitted distributions. It does not and it is problematic. If the vertical lines on the left graph are not the uncertainty, please add it. It is important.

Minor Comments

Line 207: The back trajectory model flexpart should be accompanied with proper references.

Line 262-263: Please state the magnitude of the uncertainty and do so to the rest of the article.

Line 268: Please mention what do you mean by not very intense, intense etc

Line 296 and elsewhere: When an agreement is mentioned it is proper to be followed by a indication of its robustness. Typically Pearson's R is used (R2 is certainly encouraged).

Line 769: There seems to a typo on that line.

---

## Author Response (AR1)

**Comment to the editor:**

We want to thank the two reviewers for their very useful comments.

We have added in the legend of Figure 9: "All the WALI profiles obtained between -30 min. and + 30 min. of the given times are plotted.", and we have rewritten the legend of Figure 14, which was unclear: "Each pair of graphs represents the time series of flight altitude (top) and LOAC-derived aerosol concentration for the 19 size classes (bottom), for BLPB flights from Minorca towards French coast. Colour coding is as in Fig. 7. Day-night transitions are indicated by dashed lines when appropriate."

Please note a change in the address of M. Mallet ([b]now at Centre National de Recherches Météorologiques (CNRM), UMR Météo-France-CNRS, OMP, Météo-France, Toulouse, France)

We have added several references in the introduction of our revised version to complete our literature review on the transport of large particles:

Ansmann, A., Petzold, A., Kandler, K., Tegen, I., Wendisch, M., Müller, D., Weinzierl, B., Müller, T. and Heintzenberg, J.: Saharan Mineral Dust Experiments SAMUM–1 and SAMUM–2: what have we learned?, Tellus B, 63: 403–429. doi:10.1111/j.1600-0889.2011.00555.x, 2011.

Chen, G., Ziemba, L. D., Chu, D. A., Thornhill, K. L., Schuster, G. L., Winstead, E. L., Diskin, G. S., Ferrare, R. A., Burton, S. P., Ismail, S., Kooi, S. A., Omar, A. H., Slusher, D. L., Kleb, M. M., Reid, J. S., Twohy, C. H., Zhang, H., and Anderson, B. E.: Observations of Saharan dust microphysical and optical properties from the Eastern Atlantic during NAMMA airborne field campaign, Atmos. Chem. Phys., 11, 723-740, https://doi.org/10.5194/acp-11-723-2011, 2011.

Haywood, J., Francis, P., Osborne, S., Glew, M., Loeb, N., Highwood, E., Tanré, D., Myhre, G., Formenti, P., and Hirst, E.: Radiative properties and direct radiative effect of Saharan dust measured by the C-130 aircraft during SHADE: 1. Solar spectrum, J. Geophys. Res., 108, 8577, doi:10.1029/2002JD002687, 2003.

Maring, H., Savoie, D. L., Izaguirre, M. A., Custals, L., and Reid, J. S.: Mineral dust aerosol size distribution change during atmospheric transport, J. Geophys. Res., 108, 8592, doi:10.1029/2002JD002536, 2003.

McConnell, C. L., Highwood, E. J., Coe, H., Formenti, P., Anderson, B., Osborne, S., Nava, S., Desboeufs, K., Chen, G., and Harrison, M. A. J.: Seasonal variations of the physical and optical characteristics of Saharan dust: Results from the Dust Outflow and Deposition to the Ocean (DODO) experiment, J. Geophys. Res., 113, D14S05, doi:10.1029/2007JD009606, 2008.

Prospero, J. M, and Carlson, T. N.: Saharan dust outbreaks over the tropical North Atlantic, Pure Appl. Geophys., 119, 677-691, doi:10.1007/BF00878167, 1981.

Ryder, C. L., Highwood, E. J., Lai, T. M., Sodemann, H. and Marsham, J. H.: Impact of atmospheric transport on the evolution of microphysical and optical properties of Saharan dust, Geophys. Res. Lett., 40, 2433–2438, doi:10.1002/grl.50482, 2013a.

Weinzierl, B., Sauer, D., Esselborn, M., Petzold, A., Veira, A., Rose, M., Mund, S., Wirth, M., Ansmann, A., Tesche, M., Gross, S., and Freudenthaler, V.: Microphysical and optical properties of dust and tropical biomass burning aerosol layers in the Cape Verde region—an overview of the airborne in situ and lidar measurements during SAMUM-2, Tellus B, 63, 589-618, doi:10.1111/j.1600-0889.2011.00566.x, 2011.

We have also added in Figure 6 maps of the aerosol optical depth for other dust events documented with LOAC than the mid-June event shown following the same format in Figure 4.

Our detailed response to the two reviewers' comments is following.

**Answer to Reviewer 1:**

*Reviewer: The technical descriptions of measurements are scattered throughout the*
*manuscript (ie LIDAR, airborne, satellite). It is strongly recommended that all technical details are given*
*in a separate section described as material and methods, preferably before 2.Experimental strategy*
*that could be a subsection. This will allow the reader to focus on the observations rather the technical*
*details.*
Answer: Following the reviewer comments, we have gathered the technical descriptions.
Section 2 is now subdivided in 3 sub-sections, the second one being devoted to the other instruments
and measurements used for the cross-comparisons.

*Reviewer: Line 213: "July 2.*
Answer: Correction done.

*Reviewer:* Line 244: Refer to the corresponding Figure 5.
Answer: Correction done.

*Reviewer: Line 327: In this paragraph comparison to AERONET is briefly discussed and in Figure*
*12 a single measurement is presented to support the statement that LOAC and AERONET are in very*
*good agreement. More data need to be presented, there must be several AERONET profiles available*
*to compare. These data will be useful to the AERONET community as well, since these measurements*
*may provide validation data for the inversion algorithms. Otherwise, that statement should be limited*
*to a single day that good agreement was observed.*
Answer: We have now added in Figure 12 the whole set of comparisons available between
AERONET and LOAC. We have added in the text: "The LOAC volume size distribution is compared to
that derived from the AERONET remote-sensing photometer during the 15-30 June 2013 dust events.
On average, the AERONET and LOAC data are in good agreement regarding both the overall amplitude
of the concentrations, and the position and the concentration of the coarse mode at about 3
micrometers in radius. The better agreement is on the 16 June morning; the discrepancies for the other
dates could be due to the local variability of the plume content since the LOAC and AERONET
measurements are not conducted at the same time. Nevertheless, strong discrepancies sometimes
occur for the smallest sizes (below 0.4 micrometers in radius) and for the largest sizes (above 10
micrometers in radius). The small-radius discrepancies could be due to local variability in the dust
content, like on the 27 June when AERONET retrieves a concentration increase centred on 0.25
micrometers in radius, and to respective uncertainties of both methods. On the other end of the
particle size range, AERONET retrieval is not very sensitive to the particles larger than 7 micrometers
in radius and the largest size class considered in the algorithm (15 micrometers in radius) is limited to
particles smaller than about 19.7micrometers in radius (Dubovik and King, 2000; Hashimoto et al.,
2012). Thus, LOAC could have detected large particles that were not retrievable from AERONET
observations."
*Reviewer: Line 344: This paragraph should be incorporated in Experimental strategy section,*
*you are describing once again the flight patterns here.*
Answer: We have incorporated some parts of the paragraph in the new section 2.3 and we
have rewritten the beginning of the section 4: "Figure 13 presents results from the BLPB flights
performed inside dust plumes. In particular, the 27-28 June and the 2-3 July BLPB flights were the
longest ones, with duration of about 1 day. Day-night transitions were thus encountered, leading to a
decrease in float altitude during the night of more than 100 m due to the cooling of the balloon gas
and associated loss in buoyancy, so that the night-time and daytime measurements were not
conducted in exactly the same air mass."

*Reviewer: Line 454: Evidence has to be given that this correlation exists otherwise this is rather*
*a speculation and the sentence has either to be rephrased as a hypothesis or removed.*
Answer: We have changes the text to: "In contrast, an offset increase coincident with the
increase in dust particle concentration was detected for 5 flights when crossing a dust plume, as shown
in Figure 16 . Such an offset increase was never observed outside the plumes.
Laboratory tests have shown that the LOAC electronics is indeed very sensitive to
electromagnetic fields, with an increase of the offset."
*Reviewer: Figure 5. The same height resolution should be used, it is easier for the reader to*
*compare the two plots.*
Answer: Done.
*Reviewer: Figure 11: Although the logarithmic scale shows good agreement between various*
*instruments, it would be better to present concentrations in linear scale.*
Answer: Log-scale is necessary to see both high and low concentrations and the whole size
distribution (which is always presented in log-scale in the literature). Such representation will be
impossible in a linear scale. We prefer to maintain the figure as it is.

**Answer to Reviewer 2:**

*Reviewer: My main concern is the counting statistics of the LOAC at sizes greater than 40*
*micrometers. The authors report 1e-4/ccm as the maximum observed concentration. For 1 minute*
*sampling (averaged over 6x10 sec measurements) in 1.7 Lpm means that 0.17 particles were sampled.*
*Renard et al. (2016) reports that for the smaller size classes at least 400 particles must be detected to*
*have a proper signal, which translates to 1 mV accuracy in the case of 20 mV noise. However, for higher*
*channels it is not mentioned how many particles are required for a proper signal. Furthermore, Renard*
*et al. (2016) reports a good match with a fog monitor over the size range in question, but for*
*concentrations up to 0.1/ccm (ie 3 orders of magnitude higher). As a result I encourage the authors to*
*discuss the detection limit of LOAC at sizes greater than 40 micrometers. Additionally Re[n]ard et al.*
*(2016) reports ±60 % uncertainty for concentrations smaller than 1e-2/ccm. Since the uncertainty*
*increases with decreasing concentration, can it be that in the 1e-4 range the uncertainty reaches ±100*
*%?*
Answer: In case of large particle, the proper signal translate is of about 100 mV or more, thus
well above the noise. The statistical constraints for the detection of the smallest particles do not apply
to the largest ones. The number of large particles necessary for their detection is just one per size class.
In fact, the accuracy is depending on the integration time. The given value in the Renard et al.
paper is for the basic integration time of 10s. For the results presented here, the integration time is
one minute (LDB flight) and 20 minutes (BPCL flight). Thus, the uncertainties are reduced by about 2.4
and 11, respectively. It is why the detection of the largest particle is more accurate during the BLBP
flights than during LBD flight. For a concentration of $10^{-4}$ particles cm$^{-3}$, the uncertainties can be up to
200% in case of a LDB flight but down to 25% in case of a BLBP flight.
We have modified the text in part 2.1: "In contrast, the uncertainty is up to about 60% for
concentration values smaller than $10^{-2}$ particle per cm$^3$ for a 10-s integration time."
We have added at the end of part 2.1:" The concentrations uncertainties are depending on the
integration time. Higher is the integration time, more accurate are the measured concentrations; this
is a strong constraint for the detection of the largest particles in low concentration. Typically, for
concentration lower than $10^{-4}$ particles cm$^{-3}$, the uncertainties can be high as 200% during a LDB flight,
and down to 25% for the BLBP flights with an integration time of 20 min."
We have also modified the end of part 3 to: "Since the concentration of these large particles is
low and subject to large uncertainties, the analysis of this mode from measurements during LBD flights is limited. Long duration measurements performed at constant altitude using the LOAC instrument on
BLPB gondola with much longer measurement integration time are better adapted to evaluate the
concentration of these large particles (with an accuracy down to 25%) and to discuss this third, giant
size mode."
*Reviewer: My second major point has to do with the sampling. Two sampling methods were*
*implemented, LDB and BLDB. From reading the article, I am left with the impression that both methods*
*apply a vertical (to the ground) inlet. For LDB this is understood, but for the BLDB method, this would*
*be devastating taking into account that inside dust plumes high wind speeds are frequently*
*encountered. Can you please clarify?*
Answer: The inlet was horizontal for the BLBD flight. Also, the BLBD balloon is just carried by
the wind, so that the relative velocity between the air and the inlet is close to zero. We have changed
the text in part 2.1:"The horizontal speed of a drifting balloon relatively to ambient air is supposedly
close to zero and the LOAC sampling the inlet was oriented horizontally, so that the particle sampling
efficiency should be close to 100%."
*Reviewer: Fig 14 is very puzzling to me. If I understand correctly diamonds (left graph) are the*
*measurements and the vertical line on these diamonds the uncertainty (1 std? it is not mentioned). If*
*this is the case the uncertainty line should overlap with 68% of the lines in the graph on the right. In*
*other words the measurement uncertainty should somewhat match the range of fitted distributions. It*
*does not and it is problematic. If the vertical lines on the left graph are not the uncertainty, please add*
*it. It is important.*
Answer: Indeed, the legend was not clear. The left figure is just one example. The right figure
contains all measurements. We have added the underlined words to the legend: "Left: Example of
particle volume size distribution within the desert dust plume from the BLPB flight of 19 June 2013 at
an altitude of 3.3 km, from one measurement at 12:30 UT. The black diamonds are the LOAC
measurements (with 1-σ error bars), the coloured curves represent the lognormal functions for each
of the observed modes, and the black curve represents the overall fit (sum of the 3 modes). The
geometric mean diameters (*Dm*) of the 3 modes are of 0.27, 4.6 and 34 micrometers, respectively,
with respective geometric standard deviations (σ) of 1.79, 2.14 and 1.35. Right: The 41 fitted size
distributions when the third mode was detected, retrieved from all measurements during the 19 June
BLPB flight at float altitude."
*Reviewer: Line 207: The back trajectory model flexpart should be accompanied with proper*
*references.*
Answer: We have referenced the Flexpart model with Stohl et al. (2002): Stohl, A., Eckhardt,
S., Forster, C., James, P., Spichtinger, N., and Seibert, P.: A replacement for simple back trajectory
calculations in the interpretation of atmospheric trace substance measurements, Atmos. Environ., 36,
4635–4648, doi:10.1016/S1352-2310(02)00416-8, 2002.
*Reviewer: Line 262-263: Please state the magnitude of the uncertainty and do so to the rest of*
*the article.*
Answer: The LOAC uncertainties for extinction are already given and discussed in lines 249-259
(now lines 195-205). The Lidar uncertainty is represented by the scatter of the profiles in Figure 8
*Reviewer: Line 268: Please mention what do you mean by not very intense, intense etc*
Answer: We have changed the text to: "The 28 June-2 July event was not intense in terms of
aerosol load."
*Reviewer: Line 296 and elsewhere: When an agreement is mentioned it is proper to be followed*
*by an indication of its robustness. Typically Pearson's R is used (R2 is certainly encouraged).*

Answer: We understand the reviewer concern. On the other hand, the vertical sampling of the
instruments is different, thus it is necessary to interpolate the profile before calculating the correlation
coefficients. We are not in favor of such approach, since the correlation could be dependent on how
the interpolation is performed.
*Reviewer: Line 769: There seems to a typo on that line.*
Answer: Correction done.

*__Revised version of the manuscript, changes in red__*

[revised manuscript text omitted]